# Validation of a Three-Dimensional Head and Neck Spheroid Model to Evaluate Cameras for NIR Fluorescence-Guided Cancer Surgery

**DOI:** 10.3390/ijms22041966

**Published:** 2021-02-17

**Authors:** Claire Egloff-Juras, Ilya Yakavets, Victoria Scherrer, Aurélie Francois, Lina Bezdetnaya, Henri-Pierre Lassalle, Gilles Dolivet

**Affiliations:** 1Université de Lorraine, CNRS UMR 7039, CRAN, F-54000 Nancy, France; i.yakavets@gmail.com (I.Y.); l.bolotine@nancy.unicancer.fr (L.B.); h.lassalle@nancy.unicancer.fr (H.-P.L.); g.dolivet@nancy.unicancer.fr (G.D.); 2Université de Lorraine, CHRU-Nancy, F-54000 Nancy, France; 3Institut de Cancérologie de Lorraine, F-54000 Nancy, France; victoria.scherrer2@gmail.com (V.S.); a.francois@nancy.unicancer.fr (A.F.); 4Faculté d’Odontologie de Lorraine, Université de Lorraine, 7 Avenue de la Forêt de Haye, Vandœuvre-lès-Nancy, 54500 Nancy, France

**Keywords:** indocyanine green, multicellular tumor spheroids, head and neck cancer, fluorescence-guided surgery

## Abstract

Near-infrared (NIR) fluorescence-guided surgery is an innovative technique for the real-time visualization of resection margins. The aim of this study was to develop a head and neck multicellular tumor spheroid model and to explore the possibilities offered by it for the evaluation of cameras for NIR fluorescence-guided surgery protocols. FaDu spheroids were incubated with indocyanine green (ICG) and then included in a tissue-like phantom. To assess the capability of Fluobeam® NIR camera to detect ICG in tissues, FaDu spheroids exposed to ICG were embedded in 2, 5 or 8 mm of tissue-like phantom. The fluorescence signal was significantly higher between 2, 5 and 8 mm of depth for spheroids treated with more than 5 µg/mL ICG (*p* < 0.05). The fluorescence intensity positively correlated with the size of spheroids (*p* < 0.01), while the correlation with depth in the tissue-like phantom was strongly negative (*p* < 0.001). This multicellular spheroid model embedded in a tissue-like phantom seems to be a simple and reproducible in vitro tumor model, allowing a comparison of NIR cameras. The ideal configuration seems to be 450 μm FaDu spheroids incubated for 24 h with 0.05 mg/mL of ICG, ensuring the best stability, toxicity, incorporation and signal intensity.

## 1. Introduction

Head and neck cancers are common, with more than 800,000 new cases worldwide in 2018 [1]. The treatment of choice for these tumors is surgery. The goal of these surgeries is the complete removal of the tumor with sufficient safety margins [2,3,4]. Obtaining negative excision margins is essential to achieve complete healing and avoid any recurrence of the primary tumor [5]. Positive margins may also sometimes be correlated with the development of distant metastases or a second histologically linked cancer [6]. The intraoperative evaluation of these margins is made by visual inspection and palpation, followed by the extemporaneous histological evaluation, which can sometimes take time and be inaccurate [2,7]. The emergence of new techniques, such as near-infrared (NIR) fluorescence-guided surgery (FGS), allows the surgeon to achieve a specific contrast visible to the naked eye between normal and cancer tissues. These techniques provide high-resolution images, visualization of microscopic tumor nodules, and can be tumor-specific due to targeted exogenous agents [8]. 

The advantages of NIR light (700–900 nm) include its high tissue penetration (up to several centimeters deep) as well as its low autofluorescence, thus providing sufficient contrast [9]. It is not visible to the human eye and therefore, does not change the vision of the surgical field [10]. With the rise of these techniques, a large number of NIR fluorescence cameras for FGS are emerging. Before their clinical use, these systems undergo testing in tumor cells in vitro and on animal models in vivo [8]. The transition from the in vitro to the in vivo stages is often difficult in oncology. The multicellular spheroids (MCSs) models then offer an interesting solution to this problem [11,12]. MCSs are three-dimensional aggregates of cells that mimic avascular micro-tumors or metastases [13]. In comparison to monolayer cultures, a significant advantage is that their microenvironment more closely mimics the in vivo situation, and therefore the biological behavior of the cells is closer to that encountered in vivo [14]. Thus, the application of MCSs as a tumor-tissue model could improve, standardize and simplify the assessment of imaging systems, which is essential for the success of FGS in recognizing tumors.

The aim of this study was to optimize and characterize a head and neck MCS model and to explore the possibilities offered by this model for the optimization of NIR FGS protocols.

## 2. Results

### 2.1. FaDu Spheroid Model

The 10-day growth monitoring of spheroids demonstrated that FaDu spheroids were formed at 3 days and had already reached their final size (Figure 1A). During the further cell culture, the spheroids remained stable without any noticeable size increase. The size of the spheroid is rather dependent on the quantity of seeded cells. At day 7, the seeding of 2000 cells/well resulted in 400 μm spheroids, while for 14,000 and 20,000 cells/well the size was 640 μm and 700 μm, respectively. For further experiments, we selected the seeding concentration of 5 × 10^4^ cells/mL (5000 cells per well), providing spheroids with 450 µm in diameter after 7 days of culture.

To characterize the internal arrangement of spheroids, FaDu spheroids were embedded in paraffin at 3-, 7- and 10-days post-seeding. The paraffin sections were characterized by HES and Ki67 staining (Figure 1B,C). At day 3 post-seeding, the proliferative cells were present throughout the spheroid. The necrotic core began to appear at day 7, while the proliferative cells were mainly located on the periphery of the spheroid, which can be considered a typical spheroid morphology. Further (10 days in culture), the center was massively destroyed, and the spheroid had lost stability. Thus, we selected the spheroids at day 7 of growth for further manipulations.

### 2.2. Spheroid–Indocyanine Green Interaction

The toxicity of indocyanine green (ICG) on spheroids was evaluated by clonogenic assays (Figure 2A). The clonogenic ability of cells was significantly reduced upon the increase of ICG concentration over 0.25 mg/mL. For a concentration of 0.25 mg/mL of ICG in the incubation medium, cell survival decreased by 10% compared to the control (non-treated spheroids). The calculated half maximal inhibitory concentration (IC50) was 1.53 mg/mL.

We estimated the accumulation of ICG in FaDu spheroids as a function of ICG concentration in the medium (Figure 2B), and incubation time (Figure 2C) by means of chemical extraction technique. The increase in ICG concentration in the medium resulted in higher ICG uptake in spheroids at 24 h of incubation. However, the accumulation process becomes non-linear at high ICG concentrations (over 0.25 mg/mL). FaDu spheroids become saturated by ICG and uptake seems to reach a maximum of 52 ng of ICG per spheroid. To avoid the saturation effect, we chose an ICG concentration of 0.05 mg/mL for further experiments. Figure 2C displays the amount of ICG in FaDu spheroids exposed to 0.05 mg/mL ICG at different incubation times. ICG uptake significantly increased after 1 h of incubation, and then slowly grew for several hours. After 15 h of incubation, the amount of ICG in the FaDu spheroids significantly increased compared to 6 h (31.65 ng vs. 19.98 ng per spheroids, respectively). Finally, approximately 33 ng of ICG was incorporated into each FaDu spheroid after 24 h of incubation.

To better understand ICG accumulation in FaDu spheroids, we carried out the fluorescence imaging of spheroids incubated with ICG cryo-sections (Figure 3A). At short incubation (up to 1 h), ICG preferably localized on the periphery of spheroids. According to the estimated penetration profiles, no fluorescence signal was observed at a depth above 100 µm from the periphery of spheroids at 10 and 30 min of incubation (Figure 3B). The detectable fluorescence signal of ICG became observable in the center of spheroids only from 3 h post-incubation, displaying the penetration of ICG in spheroids. For 3 h and 6 h, ICG heterogeneously distributed across the spheroids preferably in the periphery. The homogeneous distribution was reached at 15 h and did not change at 24 h, confirming the results of chemical extraction.

Finally, to analyze the mechanism of incorporation of ICG into spheroids, we incubated spheroids with 0.05 mg/mL of ICG at 4 °C for 24 h. At 4 °C, the accumulation of ICG into FaDu spheroids was 4.2 times lower than that at 37 °C (*p* = 0.029, *n* = 3), confirming the involvement of an endocytosis mechanism for ICG internalization. Of note, cell vitality after 24 h at 4 °C was similar to the control at 37 °C.

### 2.3. Spheroid Detection in Tissue-Like Phantoms

To assess the capability of the Fluobeam® NIR camera to detect ICG in tissues (Figure 4A), we embedded FaDu spheroids exposed to ICG in the tissue-like phantom (Figure 4B). Before this, we grew FaDu spheroids with different sizes of 150 µm, 250 µm and 450 µm and incubated them for 24 h with different concentrations of ICG from 1 ng/mL to 1 mg/mL. We compared the fluorescence intensity collected from ICG-stained spheroids at different depths (h = 2, 5 or 8 mm) of the phantom for each concentration of ICG (Figure 4C–E). For 450 µm spheroids, we observed a significantly higher fluorescence signal between 2, 5 or 8 mm of depth for spheroids treated with more than 5 µg/mL ICG (*p* < 0.05). Similar results were obtained for 250 µm spheroids, while for the smallest 150 µm spheroids, the concentration threshold was 50 µg/mL of ICG. To summarize the results, we plotted ICG fluorescence intensity as a function of diameter (d) of spheroids and depth (h) in the phantom for the concentration of 5 µg/mL ICG (Figure 4F). As seen from the plot, to clearly detect the spheroids, fluorescence should be more than 125 a.u. (green zone). Therefore, the spheroids smaller than 225 µm will not be detectable even at 2 mm in depth. Below this value of 125 a.u., the signal detected comes from the FaDu cells autofluorescence.

In summary, to observe a detectable signal, the ICG concentrations in the incubation medium should be higher than 5 µg/mL for large 250 µm and 450 µm spheroids, and at least 5 µg/mL for the small 150 µm spheroids. The fluorescence intensity positively correlated with the size of spheroids (Pearson coefficient = 0.39, *p* < 0.01), while the correlation with depth in the tissue-like phantom was strongly negative (Pearson coefficient = −0.51, *p* < 0.001).

## 3. Discussion

The main objective of this work was to develop an in vitro tumor model reproducing the optical properties observed in vivo. The aim of this model was to avoid the early use of in vivo experimentation for the assessment of an NIR imaging system. We developed, optimized and characterized 3D spheroids of the human head and neck squamous cell carcinoma FaDu cell line. Using the developed 3D model, we optimized the concentration and incubation time of ICG, a commonly used clinical NIR dye. Our data suggested that the optimal ICG concentration is 0.05 mg/mL for use with this spheroid-phantom model, which was applied for assessment of Fluobeam® NIR camera capabilities of NIR detection of head and neck tumors.

In vivo, solid tumors are heterogeneously exposed to oxygen and nutrients: their concentration decreases towards the center of the tumor. Conversely, metabolic waste is more concentrated in the center. It is not possible to approach these particular conditions with a two-dimensional cellular model. The use of three-dimensional models, such as spheroids, then makes sense. Spheroids reproduce the different gradients present in tumors in vivo. Like tumors, they are characterized by an exponential proliferation phase followed by a decay phase associated with cells that no longer proliferate and necrotic cells. Conversely, the two-dimensional model of monolayer cells corresponds to cells that proliferate exponentially without a decay phase [12]. Here we used FaDu cells, since they rapidly form stable, round-shape, single spheroids [15,16,17]. The spheroids produced in this work correspond well to the typical definition of spheroids made by Weiswald et al. [12]. In fact, the seeding of 5000 cells per/well results in uniform-sized spheroids with approximately 500 µm in diameter, peripheral proliferative cells and quiescent cells in the center at day 7 of culture. Additional staining of the hypoxic and necrotic zones could be interesting to continue the characterization of this model.

ICG is a FDA-approved dye and is commonly used in clinical practice [18]. Analysis of the mechanisms of incorporation revealed the existence of active transport mechanisms of ICG in spheroids. Onda et al. [19] evoked an endocytosis-like mechanism. Indeed, at 4 °C, the internalization of ICG into FaDu spheroids was 4.2 times lower than at 37 °C. We observed the incorporation of ICG after only 10 min of incubation of the spheroids with 0.05 mg/mL of ICG at 37 °C. Nevertheless, for up to 1 h of incubation, ICG is only present on the outer rim of the spheroid, up to approximately 100 μm in depth. 

It is only after 3 h of incubation that ICG heterogeneously distributed across the spheroids, but still preferably in the periphery. A homogeneous distribution was reached at 15 h and did not change at 24 h. This is in agreement with the results obtained during the quantification of ICG incorporation by chemical extraction experiments. The quantity of ICG incorporated in the spheroid increases much more slowly after 6 h of incubation and begins to stabilize after 15 h of incubation. As a conclusion, we suggest that 0.05 mg/mL of ICG is an optimal concentration for loading of spheroids since this ICG concentration is non-toxic (100% cell survival) and provides an uptake of 33 ng of the fluorophore into 450 μm FaDu spheroids at 24 h post-incubation. The time required between ICG administration and imaging of the tumor is different in vitro compared to in vivo. Indeed, in vivo, ICG is administered intravenously. If the time between injection and imaging is too short, this will result in a low signal-to-noise ratio due to high background noise. Conversely, if the time between injection and imaging is too long, there is a risk of a low signal-to-noise ratio due to signal loss [19]. Since FGS techniques are still experimental, there is no consensus on the time delay between ICG injection and imaging. In their study, Yokoyama et al. sought to determine the optimal surgical time for resection of a head and neck tumor after intravenous ICG administration [20]. To do so, they relied on the study of ICG fluorescence in healthy and tumor tissues at different post-injection times. They were able to observe fluorescence almost immediately after injection in the entire tumor region. No contrast was observed at this stage between the tumor and healthy organs. The greatest contrast between healthy tissue and tumor was observed approximately 30 to 60 min after administration of ICG. After 6 h, fluorescence emissions had decreased in the tumor but remained detectable. Twelve hours later, no contrast was observed between the tumor and healthy tissue. After taking into account the constraints of the surgical procedure, they concluded that the optimal time between the injection of ICG and imaging is between 30 min and 2 h in the case of VADS cancers (61). At this stage, the accumulation of ICG in the tumor is considered to be maximal. In our spheroid model, this level is reached for an incubation of 24 h. 

Nevertheless, it is known that the greater accumulation of ICG in tumor tissues is explained by the enhanced permeability and retention (EPR) effect [21] and therefore related to tumor vascularization. A limitation of the model presented here is thus the fact that it is avascular. An evolution of this model would be the use of a three-dimensional vascularized tumor model as proposed by Figtree et al. [22] in cardiology or Sobrino et al. [23] in oncology.

Since the main objective of this work was to develop a model enabling the comparison of different NIR fluorescence camera systems adapted to FGS, we suggest using FaDu spheroids embedded at different depths in a tissue-like phantom. This tissue-like phantom enables a simulation of the optical conditions of the tissues. The generation of such a phantom follows a simple and perfectly reproducible protocol. In this project and for the sake of convenience, the phantoms were generated in 96-well plates, however, they could be easily adapted for large volumes reproducing various shapes [24]. We have been able to detect the ICG-loaded spheroids from 150 µm to 450 µm in sizes up to 8 mm deep. For 450 and 250 µm spheroids, we observed significantly higher fluorescence signal between 2, 5 or 8 mm of depth for spheroids treated with more than 5 µg/mL ICG (concentration of the incubation solution), which corresponds to a spheroid loaded with approximately 8.5 ng of ICG (for 450 µm spheroids). For the smallest 150 µm spheroids, the ICG concentration in the incubation solution had to be 50 µg/mL to obtain a significant signal. However, at 8 mm the incubation concentration required for detection with the system used exceeded the accepted cytotoxicity with a concentration of 0.5 mg/mL (or 52 ng of ICG per spheroid). In addition, we highlighted that 450 µm FaDu spheroids appeared to reach their maximum incorporation capacity at 52 ng of ICG. Thus, the limit of detection possible with the FluoBeam® system and our model would be 8 mm deep. Similarly, in the clinical trial of Lieto et al. [25], the simultaneous use of ICG and a Fluobeam® camera allowed the removal of tumor nodules from 200 to 700 µm of diameter, undiagnosed during preoperative examinations. Thus, the results obtained using our model with the Fluobeam® camera seem perfectly consistent with the results obtained in vivo. That is why this model of FaDu spheroids included in a phantom constitutes an interesting model simulating the micro invasion of the surgical margin. In fact, at the invasive front (tumor border) of certain cancers such as most squamous cell carcinomas, invasive cells can migrate as collective groups. There is an invasion of normal tissue by compact groups or clusters of cells [26]. These cells can be a part of the tumor mass or can detach from it in the form of multicellular groups. Two fundamentally different patterns of invasive growth can be distinguished: collective cell migration and single cell migration. Collective migration is characterized by the migration of whole groups of cells interconnected by adhesion molecules and other communication junctions [27].

Bhavane et al. also highlighted the value of using phantoms to analyze the depth of detection of the ICG [28]. However, in their experiments, the ICG in free form or in liposomal form is placed purely at different depths in a phantom. The advantage of using a spheroid previously incubated with ICG and then included in a phantom is to be closer to the conditions found in vivo and therefore to be more in agreement with the situations encountered in the clinic.

## 4. Materials and Methods

### 4.1. Cell Line

The human head and neck squamous cell carcinoma cell line FaDu was obtained from the ATCC® (LGC Promochem, Molsheim, France) and regularly controlled. The cell line was maintained as a monolayer culture in Roswell Park Memorial Institute Medium (RPMI) (RPMI-1640, Invitrogen™, Carlsbad, CA, USA) supplemented with 9% heat-inactivated fetal calf serum (Sigma-Aldrich, St. Louis, MO, USA) and 2 mM L-Glutamine (Invitrogen™, Carlsbad, CA, USA). Cells were cultured in a humidified incubator at 37 °C in an atmosphere containing 5% CO_2_ and were reseeded every week.

### 4.2. Near-Infrared Camera System and Probe

The NIR camera system used in this study was Fluobeam®, developed by Fluoptics (Grenoble, France).

We chose indocyanine green (ICG) as an NIR probe due to its clinical availability. This fluorophore, with an absorbance and an emission maximum of 780 nm and 820 nm respectively, is a water-soluble molecule with high affinity for plasma proteins, lipoproteins and phospholipids. ICG was used in the form of Infracyanine® (SERB, Paris, France) with an ICG powder and a solvent (glucose water). The ICG-solvent mixture was always made at the time of experiments, protected from the light and at a temperature not exceeding 25 °C.

### 4.3. FaDu Spheroid Model

To obtain FaDu spheroids, the liquid-overlay technique was retained. After trypsinization (trypsin, GIBCO™, ThermoFisher, Waltham, MA, USA) of the monolayer-cultured FaDu cells, they were placed in different concentrations (from 0.5 × 10^4^ to 10^5^ cells/mL) in pre-coated with agarose 96-well plates (ThermoFischer, Waltham, MA, USA) and stored at 37 °C, in a humidified atmosphere and 5% CO_2_. The growth of the spheroids was monitored over 10 days with phase-contrast microscopic controls until 3 days. Spheroids were used after 7 days. The spheroids thus obtained were characterized by HES (hematoxylin eosin saffron) and Ki67 staining of sections.

### 4.4. Spheroid–Indocyanine Green Interaction

#### 4.4.1. Indocyanine Green Cytotoxicity: Clonogenic Assays

After 24 h incubation with different concentrations of ICG (0.25–5 mg/mL), spheroids were washed with DPBS (GIBCO™, ThermoFisher, Waltham, MA, USA). They were dissociated by the addition of 0.05% trypsin (GIBCO™, ThermoFisher, Waltham, MA, USA)–0.02% ethylenediaminetetraacetic acid (EDTA, GIBCO™, ThermoFisher, Waltham, MA, USA). After centrifugation and suspension of the pellet in RPMI medium, the cells were seeded at 500 cells per well. The wells were supplemented with RPMI medium and placed in the incubator for 10 days. The clones were then fixed with 70% ethanol and stained with 1% crystal violet. The colony count was done with the naked eye.

#### 4.4.2. Uptake and Chemical Extraction of ICG in Spheroids

FaDu spheroids were incubated either for 24 h at 37 °C with different concentrations of ICG (100 ng/mL–500 µg/mL) or with 0.05 mg/mL of ICG for different durations (10 min–24 h). After incubation, they underwent chemical extraction. For that, they were first washed with serum-free RPMI medium. After centrifugation at 400 g for 5 min, the supernatant was removed and replaced with an extraction solution composed of 80% absolute ethanol, 20% DMSO (dimethyl sulfoxide) and 1% acetic acid. After 15 min in an ultrasonic bath, the suspension was centrifuged for 10 min at 5000 g. The ICG fluorescence intensity was measured by spectrofluorometer (SAFAS Xenius^®^, Monaco) (λex = 690 nm, λem = 720–900 nm) and was related to fluorescence calibration curves of ICG in ethanol-DMSO-acetic acid. The obtained ICG concentration was normalized to the number of spheroids.

To analyze the mechanism of incorporation of ICG into spheroids, the same protocol was implemented by incubating spheroids for 24 h with 0.05 mg/mL of ICG at 37 °C or at 4 °C. Before that, control of the maintenance of the cellular vitality after 24 h at 4 °C was carried out.

#### 4.4.3. Diffusion of ICG into Spheroids

FaDu spheroids were incubated with 0.05 mg/mL of ICG for different durations (10 min–24 h) at 37 °C. After incubation, they were washed with PBS and embedded into the resin matrix (Shandon^TM^ Cryomatrix^TM^, ThermoFisher, Waltham, MA, USA). Then, 10 µm thick frozen sections were made. The sections were observed using an upright epifluorescence microscope (Provis, AX-70, Olympus, Rungis, France) using the filter set at 770 ± 20 nm excitation associated with an 801 nm dichroic mirror and an 830 ± 20 nm long-pass emission filter for fluorescence measurements.

Fluorescence images were analyzed with ImageJ (NIH, Bethesda, MD, USA) software and with the macros proposed by Yakavets et al. [29] to estimate the penetration profile of ICG in the spheroids. The spheroid area was divided into 100 concentric rims, with the diameter decreasing in a linear way. After that, we calculated the mean intensity of pixels in each rim. The final profiles were plotted as mean ± standard deviation from different cryo-sections (*n* = 4) [30].

### 4.5. Spheroid Detection in Tissue-Like Phantoms

In order to mimic the detection sensitivity of ICG in biological tissues, a tissue-like phantom mimicking the optical properties of tissue in the NIR was produced. The phantom used was the one proposed by De Grand et al. [24]. This phantom consists of a mixture of gelatin, intralipid and bovine hemoglobin mixed in a Tris buffer (Tris, NaCl, sodium azide, H_2_O).First, a constant phantom thickness (1 mm, which corresponds to 32 μL of phantom) was deposited in 96-well plates. Then, FaDu spheroids (150 to 450 μm in diameter) incubated for 24 h with ICG (1 × 10^−6^ to 1 mg/mL) were placed on the phantom and then covered with 2, 5 or 8 mm of the phantom. Thus, the whole spheroid was in an identical environment. The reading of the 96-well plates placed on black support was made with a Fluobeam^®^ camera. ICG was excited with a wavelength of 690 nm and the fluorescence emitted was detected by the Fluobeam^®^ imager between 700 and 850 nm, with an exposure time of 150 ms.

All acquired images were analyzed by annotating regions of interest (ROI) and using the software ImageJ. The background ROI was positioned at a location of homogeneous intensity without spheroids. The background signal from phantom-only wells was subtracted for each sample.

### 4.6. Statistical Analysis

All results were expressed as mean values ± standard deviation (SD). One-way analysis of variance (ANOVA) followed by Tukey’s multiple comparisons test was used to compare the means between more than two means. Students’ *t*-test was used to establish a comparison between two means. The level of significance was set at 0.05. Statistical analyses were performed using the Origin software (OriginLab, MA, USA).

## 5. Conclusions

NIR FGS is an innovative and interesting technique for the real-time visualization of resection margins limiting the risk of recurrence and tissue loss. Different NIR fluorescence cameras are available and it seems essential to test their limits to control their use. That is why we have developed and optimized a simple and reproducible in vitro tumor model, allowing a simple comparison of these different systems: a multicellular spheroid model embedded in a tissue-like phantom. The ideal seems to be to use 450 μm FaDu spheroids incubated for 24 h with 0.05 mg/mL of ICG ensuring respectively the best stability, toxicity, incorporation and signal intensity. Now it would be interesting to test this model with other camera systems and to develop other phantom morphologies (closer to the human anatomy).

## Figures and Tables

**Figure 1 ijms-22-01966-f001:**
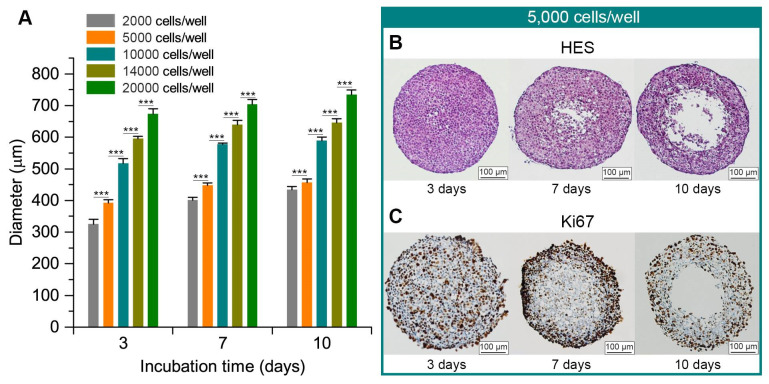
(**A**) Spheroid growth kinetics initiated with different cell concentrations. FaDu cells were placed in different concentrations (from 0.5 × 10^4^ to 10^5^ cells/mL) in agarose pre-coated 96-well plates and then stored at 37 °C, in a humidified atmosphere with 5% CO_2_. Growth of the spheroids was monitored from day 3 to day 10 with phase-contrast microscopic controls. *** *p* < 0.00001. (**B**,**C**) Paraffin sections of 3-, 7- and 10-day spheroids initiated from 5000 cells/well. Paraffin sections of 3-, 7- and 10-day spheroids initiated from 5000 cells/well were stained with Hematoxylin-Eosin-Safran (HES) (**B**) and Ki67 (**C**). The scale bar represents 50 µm.

**Figure 2 ijms-22-01966-f002:**
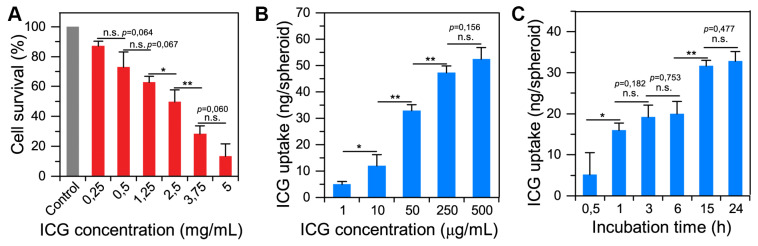
(**A**) Cytotoxicity of ICG on FaDu spheroids evaluated by clonogenic assays. After 24 h incubation with different concentrations of indocyanine green (0.25–5 mg/mL), cells were dissociated and seeded at 500 cells per well in plates. The wells were supplemented with RPMI medium and placed in the incubator for 10 days. The clones were then fixed with 70% ethanol and then stained with 1% crystal violet. The colony count was done with the naked eye. The results are presented as mean ± SD with *n* = 4. A * indicates a significant difference with *p* < 0.05. ** indicates a significant difference with *p* < 0.01. (**B**) Incorporation of ICG in FaDu spheroids after 24 h of incubation at different concentrations. FaDu spheroids were incubated 24 h with different concentrations of indocyanine green (1 µg/mL–500 µg/mL). After extraction, the fluorescence intensity was measured by a spectrofluorometer and was related to fluorescence calibration curves of ICG. The obtained ICG concentration was normalized to the number of spheroids. The results are presented as mean ± SD with *n* = 4. (**C**) Incorporation of ICG in FaDu spheroids after different incubation time. FaDu spheroids were incubated with 0.05 mg/mL of ICG for different durations (0.5–24 h). After extraction, the fluorescence intensity was measured by spectrofluorometer and was related to fluorescence calibration curves of ICG. The obtained ICG concentration was normalized to the number of spheroids. The results are presented as mean ± SD with *n* = 4.

**Figure 3 ijms-22-01966-f003:**
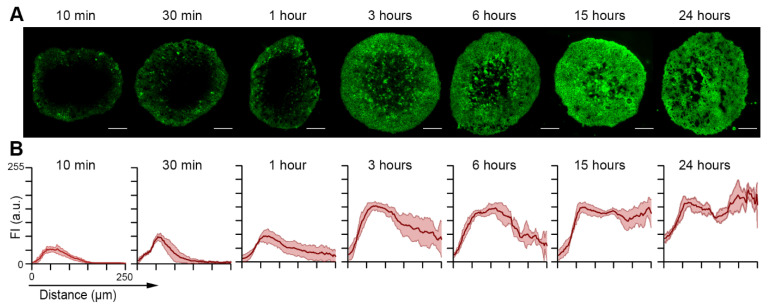
Distribution of ICG in FaDu spheroids after different incubation times. (**A**) Typical fluorescence images of ICG fluorescence in FaDu spheroids after different incubation times with 0.05 mg/mL of ICG. All scales bars represent 100 μm. (**B**) Penetration profiles of ICG in FaDu spheroids after the same incubation time. ICG concentration was 0.05 mg/mL. The data are presented as mean ± SD obtained from *n* = 4 spheroids.

**Figure 4 ijms-22-01966-f004:**
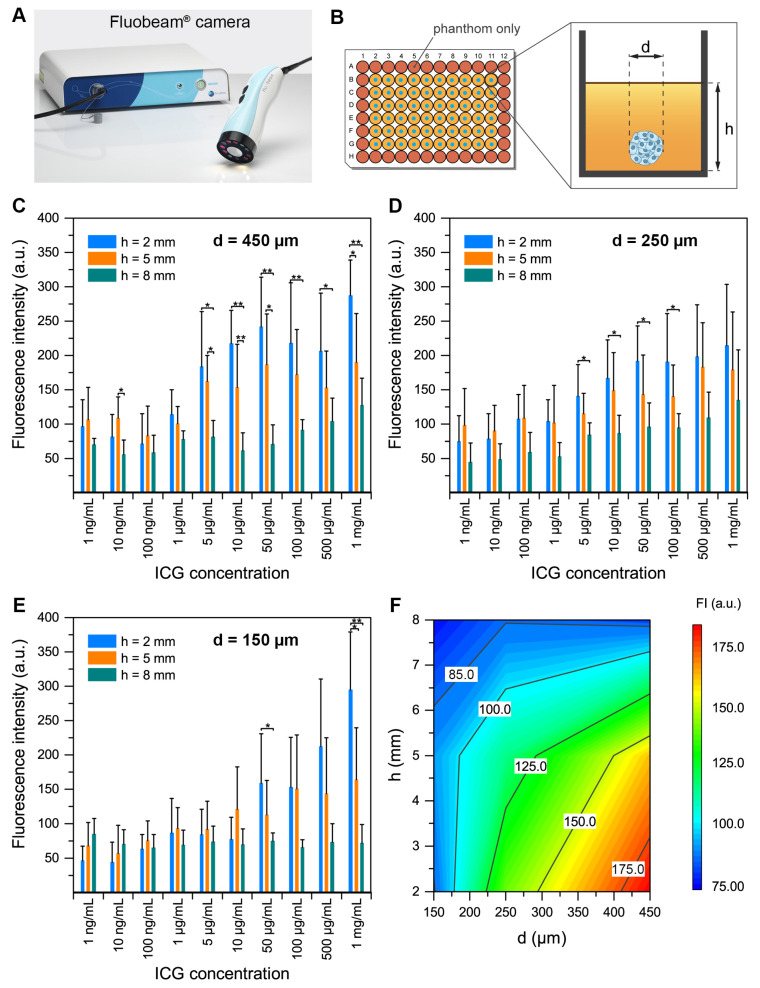
Detection in tissue-like phantoms of spheroids incubated for 24 h with ICG using the Fluobeam^®^ camera. (**A**) Photo of the Fluobeam^®^ camera. (**B**) FaDu spheroids (150 to 450 µm in diameter (d)) incubated for 24 h with ICG (1 × 10^−6^ to 1 mg/mL) were placed onto the phantom and then covered with different heights (h) of the phantom. ICG was excited with a wavelength of 690 nm and the fluorescence emitted was detected by the Fluobeam® imager between 700 and 850 nm, with an exposure time of 150 ms. (**C**–**E**) Detection of 450, 250 and 150 μm FaDu spheroids in 2, 5 or 8 mm of tissue-like phantom. The results are presented as mean ± SD with *n* = 4. ANOVA + Tukey post-hoc test. * *p* < 0.05; ** *p* < 0.01. The background signal from “phantom only” wells was subtracted for each sample. (**F**) Plot of ICG fluorescence intensity as a function of diameter (d) of spheroids and depth (h) in the phantom for a concentration of 5 µg/mL of ICG.

## Data Availability

The data that support the findings of this study are available from the corresponding author upon reasonable request.

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
