# Peer review of "Validation of a Three-Dimensional Head and Neck Spheroid Model to Evaluate Cameras for NIR Fluorescence-Guided Cancer Surgery"

_ijms, 2021, doi:10.3390/ijms22041966_

Round 1

Reviewer 1 Report

Overall the manuscript is interesting and the technique would be a valid way of evaluating cameras for NIR fluorescence guided cancer surgery.

Questions, suggestions, and corrections are contained in the attached document

Regards

Reviewer 2 Report

This study characterized and optimized a head and neck MCS model and explore the possibilities offered by this model for the optimization of the NIR FGS protocol, which may pave the way to develop a new detection strategy for cancer treatment. While this topic is interesting, this study cannot yield meaningful results by the lack of data rigor. Many substantial flaws in the design and conduction are identified as follows.

  1. The data generated from one cell line cannot produce broad applicability. At least one more HNSCC cell line should be added to this study to validate the results. The data from two or three cell lines may have a huge difference, but this could provide strong rationales for building a more accurate MCS model which should enhance the significance of the study.
  2. In this study, FaDu spheroid model was only generated in agarose gels. Did the authors try to create FaDu spheroid using other materials? I will encourage authors to perform the same assays in other types of 3D spheroid.
  3. Some statements are not correct, for example, Head and neck cancers rank sixth among the most common cancers is no longer the fact.
  4. I will encourage authors to discuss more in detail the mechanism that links ICG administration with head and neck cancer cells in your model.
  5. The authors should minimize the use of abbreviations in the abstract.

Reviewer 3 Report

This is an interesting study about a three-dimensional head and neck spheroid model used to evaluate cameras for NIR fluorescence-guided cancer surgery. The authors analyzed FaDu spheroids were incubated with indocyanine green (ICG) and then included in a tissue-like phantom to assess the capability of Fluobeam® NIR camera to detect ICG in tissues.

The paper is well written. However, some issues remain.

All the acronyms should be defined at their first appearance also in the abstract.

In the Introduction, the authors stated that positive margins may be correlated with the development of distant metastases or a second histologically linked cancer. Actually, positive margins are mainly related to recurrence. Please implement and correct the sentence.

Please add more data from literature on fluorescence-guided surgery and in vivo results in the Discussion section.

Reviewer 4 Report

Authors utilized a  multicellular spheroid model of FaDu cells embedded in a tissue-like phantom as a simple and reproducible in vitro tumor model, allowing a comparison of NIR cameras. The article is based on the requirement of getting tumor-free margins in surgical treatment of head and neck cancer. The tumor-free nature of remaining tissue after resection is a major determinant for clinical progress of the disease as well as for patient survival.  Utilization of fluorescence techniques for the evaluation of resection margins will give more accuracy compared to visual observation.

This Reviewer agrees with the authors that a three-dimensional spheroid system would provide a better test model for testing the NIR-cameras than a 2D standard cell culture.

Reviewer Questions

What is the main aim of using  indocyanine green (ICG) staining and detection in the model spheroid and finally in the tissue during surgery? In a short staining time (10 minutes would be realistic in surgical setting) to detect accumulated proliferaring cells in a background of necrosis and normal eventually not proliferating normal tissue and in stroma?

What are the main outcomes from the used models, which would be relevant for the surgical setup, in terms of staining condition, use conditions and limitations? 

Round 2

Reviewer 2 Report

Although the authors didn’t address all my concerns, the quality of this revision has been improved. I have no further questions.